# Audiological Risk Factors, Referral Rates and Dropouts: 9 Years of Universal Newborn Hearing Screening in North Sardinia

**DOI:** 10.3390/children9091362

**Published:** 2022-09-07

**Authors:** Laura Maria De Luca, Rita Malesci, Roberto Gallus, Andrea Melis, Sara Palmas, Emilia Degni, Claudia Crescio, Maria Lucia Piras, Maria Francesca Arca Sedda, Giovanna Maria Canu, Davide Rizzo, Mauro Giorgio Olzai, Salvatore Dessole, Giovanni Sotgiu, Anna Rita Fetoni, Francesco Bussu

**Affiliations:** 1Otolaryngology Division, Azienda Ospedaliero Universitaria, 07100 Sassari, Italy; 2Audiology Section, Department of Neuroscience, Reproductive Sciences and Dentistry, Federico II University, 80138 Naples, Italy; 3Otolaryngology, Mater Olbia Hospital, 07026 Olbia, Italy; 4Otolaryngology Division, Department of Medical, Surgical and Experimental Science, University of Sassari, 07100 Sassari, Italy; 5Neonatology and Neonatal Intensive Care Unit, Azienda Ospedaliero Universitaria, 07100 Sassari, Italy; 6Gynecologic and Obstetric Clinic, Department of Surgical, Microsurgical and Medical Sciences, University of Sassari, 07100 Sassari, Italy; 7Clinical Epidemiology and Medical Statistics Unit, Department of Medical, Surgical and Experimental Sciences, University of Sassari, 07100 Sassari, Italy

**Keywords:** otoacoustic emissions, hearing loss, unilateral hearing loss, neonatal screening, risk factors

## Abstract

Background: Objectives of the present work were to analyze the prevalence of hearing loss in our population of screened newborns during the first 9 years of the universal newborn hearing screening (UNHS) program at University Hospital Sassari (Italy) (AOU Sassari), to analyze the risk factors involved, and to analyze our effectiveness in terms of referral rates and dropout rates. Methods: Monocentric retrospective study whose target population included all the newborns born or referred to our hospital between 2011 and 2019. Results: From 2011 to 2019, a total of 11,688 babies were enrolled in our screening program. In total, 3.9‰ of wellborn babies and 3.58% of neonatal intensive care unit (NICU) babies had some degree of hearing loss. The most frequently observed risk factors among non-NICU babies were family history of hearing loss (3.34%) and craniofacial anomalies (0.16%), among NICU babies were low birth weight (54.91%) and prematurity (24.33%). In the multivariate analysis, family history of hearing loss (*p* < 0.001), NICU (*p* < 0.001), craniofacial anomalies (*p* < 0.001), low birth weight (<1500 g) (*p* = 0.04) and HIV (*p* = 0.03) were confirmed as risk factors. Conclusions: Our data are largely consistent with the literature and most results were expected, one relevant exception being the possible role of NICU as a confounding factor and the limited number of risk factors confirmed in the multivariate analysis.

## 1. Introduction

Permanent congenital hearing loss is an epidemiologically relevant condition with a rate among newborns of 1 to 3 per 1000 live births. While in a relevant number of cases a definite cause cannot be established, known etiologies and risk factors include genetic factors (syndromic and non-syndromic), congenital infections, ototoxic medications, hyperbilirubinemia, premature birth, low birth weight, admission to a neonatal intensive care unit, birth injuries, drug and alcohol use during pregnancy, maternal diabetes, preeclampsia, and anoxia [1].

Hearing impairment is especially concerning at an early age because it has been associated with difficulties in the development of verbal communication, behavioral disorders, and poor adaptive skills. Early diagnosis and habilitation (within 6 months of age) are associated with better communication skills, which impact cognitive and socioemotional behavior [2].

Universal Newborn Hearing Screening programs’ (UNHS) main objective—as defined by the Joint Committee on Infant Hearing (JCIH)—is the detection of permanent sensory or conductive hearing loss. These programs are being implemented as a standard of care across the world, as recommended both by the JCIH and since 2008 by the U.S. Preventive Services Task Force [3]. They usually comprise a first stage of screening with objective tests (transient otoacoustic emissions—TEOAEs, distortion product otoacoustic emissions—DPOAEs and/or automatic auditory brainstem responses—AABRs) and a diagnostic confirmation with auditory brainstem responses (ABRs).

As defined by the American Academy of Pediatrics, the referral standard from the first stage should be <4% and the dropout rate at follow-up should be <5%, but this rate is typically higher in real world settings [4]. With the implementation of UNHS programs, a steady flow of data coming from some of them has been published, allowing for a deeper understanding of the expected and unexpected associated problems, such as costs [5], settings, technologies involved [6], referral rates [7], dropout rates [8], type of operators involved [9], and risk factors requiring special consideration [10].

Sardinia is the second biggest island of the Mediterranean Sea, with a low population density. Its historical, social, and genetic peculiarities derive from a condition of relative and unusual isolation in the Western world, which also previously worked as a real-world lab with unexpected and relevant findings [11].

The primary objective of the present work was to analyze the prevalence of hearing loss in our population of screened newborns during the first 9 years of the UNHS program at AOU Sassari. Secondary objectives were to analyze the risk factors involved in hearing loss in our UNHS population, to analyze our effectiveness in terms of referral rates and dropout rates both overall and on a yearly basis.

## 2. Materials and Methods

### 2.1. Outcomes and Population

The predefined outcomes of the study were the prevalence of hearing loss in our newborn population, the prevalence and type of risk factors, the referral rates, and the dropout rates. Our service is located inside the largest hospital in the island and is responsible for the UNHS for all the babies born at our facility and at the other hospitals in Northern Sardinia (Sassari Province) with a birth center inside, namely Giovanni Paolo II Hospital in Olbia, Ospedale Paolo Merlo Hospital in La Maddalena, Paolo Dettori Hospital in Tempio Pausania, Antonio Segni Hospital in Ozieri, Casa di cura Policlinico sassarese in Sassari, Ospedale Civile in Alghero. According to the last data available (2019) published by the Italian National Institute of Statistics (ISTAT), these hospitals serve a population of 330,211 individuals, with a birth rate of 5.8‰ (2020). As a second level center, our duty is to perform a UNHS to all newborns, and to provide a final diagnosis on site. Considering a known prevalence of newborn hearing loss of 1–3/1000 live births and the birth rate of our area, to calculate its prevalence in our population (main outcome), our goal was to enroll at least 494 newborns per year (CL 99%, CI 5%). In order to perform our analysis, clinical and screening data from electronic records were retrospectively collected from the beginning of the program until the end of 2019. The inclusion criterion for newborns included being born in our serviced area. All data were anonymized during acquisition in electronic datasheets. 

### 2.2. Our Screening Program

At AOU Sassari, the UNHS program started in 2011. NICU (Neonatal Intensive Care Unit) and non-NICU babies, born at AOU Sassari or referred from nearby centers lacking an onsite screening program, are screened with a multi-step protocol. The first stage involves the execution of TEOAEs before discharge (within 3 days) or at first contact in case of referred babies. Tests are performed in a noiseless setting, at the Department of Neonatology or inside the NICU, by a team of expert audiology technicians. During this phase, relevant prenatal and perinatal audiological risk factors are investigated and recorded. The two possible results of this first stage are “Pass” or “Refer” [9]. Non-NICU babies without known audiological risk factors and “Pass” result at the universal screening exit the screening program. Newborns with a bilateral or unilateral “Refer” result are referred to the second stage, which involves a re-test with TEOAEs at about 30 days of life. Parents of babies referred to re-test are counseled, in order to manage parents’ anxiety and minimize the dropout rate. When a unilateral or bilateral “Refer” is recorded again, a diagnostic ABR is performed by the third month. All NICU newborns and non-NICU newborns with known audiological risk factors (as defined by JCIH [3]) are referred to diagnostic ABR, independently of the outcomes of the first and second stage of screening (Figure 1).

Two different screening instruments were used during the whole period (2011–2019): Madsen AccuScreen Pro TA Hearing TEOAE and—since the beginning of 2019 only for NICU babies—Madsen AccuScreen Type 1077 TE/DP (for technical information refer to Manufacturer’s website: https://hearing-balance.natus.com/ accessed on 10 January 2022; Pleasanton, CA, USA). Both Madsen Accuscreen^®^ models allow TEOAEs recording and evaluation through noise-weighted averaging and counting of significant signal peaks. Tests are considered not passed (positive) when at least one ear has a negative result (i.e., “Refer”).

At our institution, ABRs are recorded in a silent room cabin with an Amplaid MK12 (Amplaid, Milan, Italy) by experienced audiological technicians and interpreted by an experienced audiologist. Before testing, babies are evaluated by an otolaryngologist and parents are counseled again also on the practical execution and rationale of the diagnostic test. Any baby with a hearing threshold higher than 25 dB in at least one ear at ABR is considered affected by hearing loss. Babies with a confirmed hearing loss are then referred to an evaluation with an audiologyst that includes a tympanogram. If middle ear effusion is suspected, the baby is treated as per common practice before the next evaluation. ABR tests are repeated two more times within the 6th month of life to obtain the most accurate audiological diagnosis and properly recommend early habilitation or follow-up, as indicated. Babies that have normal results at the follow-up ABRs are discharged or further audiological follow-up is indicated according to the known risk factors.

### 2.3. Data Analysis

NICU and non-NICU babies represent two markedly different populations in terms of hearing loss prevalence and number and types of risk factors involved. Therefore, data coming from these two arms were collected and analyzed both together and separately. Quantitative data collected for the analysis included the prevalence of referred babies, prevalence of dropouts, prevalence of hearing loss, and total number of risk factors. Qualitative variables included presence of hearing loss, presence of risk factors included in the JCIH of 2007, and additional clinical parameters recorded during our screening program (maternal diabetes and any maternal viral positivity other than the TORCH complex that includes toxoplasmosis, syphilis, hepatitis B, rubella, cytomegalovirus, and herpes simplex). Statistical analysis was performed with JASP (JASP Team (2021). JASP (Version 0.16) [Computer software], copyright 2013–2021 University of Amsterdam, Netherlands). A descriptive analysis of our data was performed to find both prevalence and distribution of the recorded variables. Pearson’s chi-squared test was used to assess whether there was a statistically significant difference between the frequencies of known and potential risk factors among healthy and hearing-impaired newborns. A logistic regression was performed to ascertain the effects of the total number of concomitant risk factors on the likelihood that newborns had a hearing loss. Only risk factors that were originally recorded during the screening phase and the results of the last diagnostic ABR were included in the analysis, as clinical and audiological follow-up data were often not available or reliable enough. For statistical purposes the hearing loss degree and the hearing loss status were assessed on the last available ABR, if less than three were available. Temporary cases with normal results at the follow-up ABRs were not included in the analysis. A correlation matrix of the dropout and referral rate at each stage and different years of screening was performed, and Pearson’s r and *p*-values were calculated. Significance was set at *p* ≤ 0.05 for all statistical tests. Confidence interval for all data shown is 95%.

## 3. Results

From 2011 to 2019, a total of 11,688 babies (2737 NICU and 8951 non-NICU) were enrolled in our screening program. In total, 35 of the 8951 of non-NICU babies (3.9‰) and 98 of the 2737 NICU babies (3.58%) had some degree of hearing loss, as confirmed with diagnostic ABR. Bilateral cases were 27 among non-NICU babies (27/35, 77.14%) and 57 among NICU cases (57/98, 58.16%). Cases associated with risk factors were 14 among non-NICU babies (14/35, 40%) and 98 among NICU cases (98/98, 100%). Two babies among non-NICU babies (2/8951, 0.02%) and two among NICU babies (2/2737, 0.07%) had profound bilateral hearing loss.

Of the four babies with profound bilateral hearing loss, two received an indication for cochlear implantation. For a third one, the diagnosis was corrected from profound to moderate hearing loss during follow-up (potential late partial maturation of the auditory path [12]). Finally, the last one had a severe cognitive impairment due to extensive cerebral damage (during delivery) and a cochlear implant was not indicated. Among the babies with moderate-severe hearing impairment, one coming from NICU was habilitated through a bone-anchored hearing aid (BAHA^®^, Cochlear Limited, Sydney, Australia), whereas the others through traditional external hearing aids.

Refer and drop-out rates by year are shown in Table 1. Overall, 0.61% of the screened non-NICU newborns (55/8951) and 2.41% of NICU newborns (66/2737) were prescribed an ABR due to a positive result after TEOAEs second stage. For the others, a diagnostic ABR was indicated due to the presence of risk factors, or due to their NICU stay, in agreement with our policy. In total, 52.04% (51/98) of NICU newborns and 14.28% (5/35) of non-NICU newborns with hearing impairment had passed the first stage of screening and were referred to ABR due to risk factors. The referral rate of the first screening level (TEOAEs) in the NICU population decreased steadily and significantly through the years with a significant negative correlation with time (Pearson’s r = −0.849, *p*-value = 0.002).

Descriptive statistics including sex, prevalence of each recorded risk factor and audiological features are shown in Table 2. In total, 48.92% of non-NICU babies (4379/8951) and 45.88% of NICU babies (1256/2737) were female. The most frequently observed risk factors among non-NICU babies were family history of hearing loss (299/8951, 3.34%), craniofacial anomalies (15/8951, 0.16%), syndromes (14/8951, 0.15%), and TORCH complex (12/8951, 0.13%). Among NICU babies, the most frequent risk factors were low birth weight (1503/2737, 54.91%), prematurity (666/2737, 24.33%), mechanical ventilation (661/2737, 24.15%), and hyperbilirubinemia (475/2737, 17.35%). The total number of concomitant risk factors between NICU and non-NICU newborn was statistically different (*p*-value < 0.001), with a higher number of concomitant risk factors in NICU newborns.

The results of the comparison of frequency distribution of risk factors among healthy and hearing-impaired babies, both for the whole series and for NICU and non-NICU newborns separately, are displayed in Table 3.

The results of the univariate and multivariate logistic regression for the whole series are displayed in Table 4. In the multivariate analysis, family history of hearing loss (OR 17.55; *p*-value *p* < 0.001), NICU (OR 11.45; *p*-value *p* < 0.001), craniofacial anomalies (OR 9.62; *p*-value *p* < 0.001), low birth weight (<1500 g) (OR 3.19; *p*-value = 0.04), and HIV (OR 13.20; *p*-value = 0.03) were confirmed as risk factors.

In total, 41.35% (55/133) of hearing-impaired babies were bilateral pass at the first stage of screening, and received a diagnostic ABR due to known risk factors. Risk factors among these newborns were NICU (50/55, 90.90%), low birth weight (<1500 g) (19/55, 34.54%), prematurity (12/55, 21.81%), mechanical ventilation for at least 5 days (11/55, 20%), hyperbilirubinemia (6/55, 10.9%), family history of hearing loss (5/55, 9.09%), and syndromes (1/55, 1.81%). None of them were profound or severe bilateral cases, and 25/55 (45.45%) were unilateral hearing losses. In total, 30.90% (17/55) of these cases had only one risk factor involved; in 12 cases (12/55, 21,81%) NICU and in 5 cases (5/55, 9.09%) familiarity. Moreover, 40% of non-NICU babies (14/35) with hearing loss did not have any known risk factor.

## 4. Discussion

Newborn hearing screening programs are a precious health and social tool that allow early detection and habilitation of hearing-impaired newborns. The data flow coming from programs that keep electronic or paper records has proven to be invaluable in understanding how different approaches impact the UNHS programs, in highlighting critical aspects, and in guiding preventive measures. Our screening program follows a fairly common protocol that involves TEOAEs as the test of choice for the first and second stage of screening and ABR for the definitive diagnosis. TEOAEs and aABRs are the most employed technologies in newborn hearing screening, as they are fast, reliable, and validated. They are both sensitive (85–100%) and specific (91–95%). Nonetheless, depending on the protocol applied, a not negligible number of referrals to audiological centers may be due to false positive results. The combination of TEOAEs and aABRs in the screening have been reported to reduce the referral rate [13]. Factors negatively impacting performance of such screening tools include a noisy environment, collapse or presence of debris inside the external auditory canal and mucus in the middle ear, and usually resolve within a few hours or days [4,14]. Therefore, multi-phased protocols such as ours appear rational and are widely adopted. One important limitation of TEOAEs is that they only test cochlear function, missing neural dysfunction caused by some risk factors. That is one of the reasons we refer (as many other authors) newborns with risk factors to ABR testing as a standard procedure. However, some cases of auditory neuropathy can still be missed and diagnosed in an untimely manner.

Referring newborns with risk factors to ABR testing independently from their first stage TEOAEs result has proven to be of utmost importance in our experience, as 52.04% of NICU hearing-impaired newborns and 14.28% of non-NICU ones would have been missed if not for such a practice, consistently with other reports [15]. More in detail, 5 non-NICU newborns that would have been missed had familiarity as the only risk factor. ABR, and in particular aABR if available, are recommended by the JCIH in the NICU population to avoid missing babies affected by auditory neuropathy spectrum disorder (ANSD), a condition that is particularly frequent in this setting [3]. Our results in terms of referral rate are consistent with the range reported in the literature after the first stage with 8.40% of non-NICU and 7.23% of NICU babies referred to the second stage [13]. Percentages are higher than the target indicated by the American Academy of Pediatrics and the JCIH when only the first stage is considered, but within the limits when both stages are considered (0.61% non-NICU newborns, 2.41% NICU) [16]. Interestingly, our first-stage referral rates for NICU babies have shown a statistically significant drop through the years. After an internal review, we concluded that this difference is probably related to a change in the management of the screening program that occurred at the beginning of 2015, when a reorganization of the shifts of the technical personnel resulted in increased time dedicated to the screening tests. It is important to acknowledge that such referral rates are still too high and are known to lead to a loss of efficiency and parental anxiety, an aspect that we try to address with proper counseling, including an explanation of the test’s meaning and of the high probability of a false positive result.

One of the most discussed aspects and biggest issues of all UNHS programs is the dropout rate. “Lost to follow-up” rates usually range from 3.7 to 65% [17], mainly due to a lack of parental involvement [18]. Our screening program is no exception with a dropout rate of 30.07% after the first round of TEOAEs and 30.27% of newborns missing the planned ABR, despite all the efforts in terms of parental counseling and recall campaigns. It is important to note that multi-step programs such as ours, while effective in reducing the referral rates and the burden in terms of diagnostic ABRs, lead to higher rates of newborns lost to follow-up [19].

Clinical features showed a higher prevalence of risk factors and a greater number of simultaneous risk factors among NICU babies, as expected. The distribution of risk factors we pointed out might seem atypical, with a low number of genetic anomalies [20]. However, this is actually due to the fact that genetic testing results usually come later during the full diagnostic workup, and late results coming from the post screening follow-up were not recorded in our study.

Hearing loss prevalence was within expectations, albeit in the upper limit of the reported range [21]. In total, 0.39% of non-NICU and 3.58% of NICU babies had some degree of hearing loss, while, respectively, only 0.04% and 0.07% had profound hearing losses potentially amenable to cochlear implantation surgery. On average, slightly more than one newborn every 1000 (1.1) was diagnosed with some degree of hearing loss. Our rather high prevalence of HL could be explained by our extremely strict internal protocol that includes in the definition all hearing losses above 25 dB. Most of the impaired babies had bilateral hearing loss (77.14% non-NICU and 58.16% NICU babies). Interestingly, separating hearing losses according to their degree (based on the American Academy of Speech Language Pathology—ASHA classification of hearing loss) a non “normal” distribution and a lack of babies with “severe” hearing loss (71–90 dB) could be noticed (only one for each group).

In total, 41.35% of hearing-impaired babies would have been missed if not referred to the diagnostic ABR due to risk factors, being bilateral pass at the first stage of screening, thus supporting the indication of a diagnostic ABR in newborns with risk factors even in case of a negative screening, as advocated by some authors [1]. However, most of them were mild and moderate bilateral or unilateral cases. The number of NICU babies that passed the first stage of screening but had pathological diagnostic ABR is particularly high, and higher than the non-NICU counterpart. A possible interpretation might be that these babies had an etiologic factor that caused a delayed hearing loss, or the etiologic factor itself occurred in the time frame between the TEOAEs and the diagnostic ABR, but one of the main factors at play, and the reason JCIH recommends ABR testing for NICU babies as mentioned above, is that ANSD is particularly frequent in this population and is missed by TEOAEs testing. Conversely, 40% of hearing-impaired non-NICU babies did not have any risk factor, and thus, would have been missed by a non-universal program, clearly supporting UNHS programs. Thus, only the combination of a UNHS and of a diagnostic ABR in newborns with risk factors could have allowed the detection of all hearing-impaired newborns in our series.

Our analysis on the whole series showed that most of the currently known risk factors had a statistically significant higher frequency in hearing-impaired newborns. Such risk factors included: family history of hearing loss, low birth weight (<1500 g), hyperbilirubinemia, low Apgar score [4,10], defined syndromes (CHARGE, Down, Goldenhar, Usher, Waardenburg), prematurity, NICU stay, mechanical ventilation for at least 5 days, and craniofacial anomalies. Additionally, the total number of concomitant risk factors and HIV, already noted to be associated with hearing loss by others [22,23], had the same behavior in our series. Some expected risk factors did not reach statistical significance, including TORCH complex and ototoxic medications. A possible explanation could be that sometimes they are associated with a late development of hearing loss [4], and therefore, were missed in our study (i.e., CMV). In addition, hearing losses associated with known risk factors may be missed due to an isolated involvement of high frequencies not discovered until later in life [24]. Moreover, treatment for congenital toxoplasmosis and rubella vaccine have significantly reduced the risk of hearing loss for such causes [10]. Interestingly, when controlling the confounding influence of the risk factor “NICU stay” by restriction, hence repeating the statistical analysis separately on non-NICU and NICU cases, statistical significance for some items was lost on NICU newborns, including family history of hearing loss (that retained significance on non-NICU babies), low birth weight (<1500 g), hyperbilirubinemia, prematurity, and mechanical ventilation. The only possible explanation for such behavior is that NICU stay in our series acted as a confounding factor and is probably one of the most relevant risk factors at play, minimizing the relevance of other well-known risk factors. Such relevance has been further confirmed by our multivariate analysis, which confirms only the role of familiarity, NICU stay, craniofacial anomalies, low birth weight, and HIV. While the role of familiarity, craniofacial anomalies, and low birth weight is easily justified, the role of NICU as a standalone risk factor is less clear. One possible explanation of the role of NICU might be that the higher number of concomitant risk factors in these newborns might play a role and determine a higher risk of developing hearing loss, as suggested by the increased likelihood of hearing loss in babies with a higher number of concomitant risk factors in our series and the higher frequency of newborns with concomitant risk factors in the NICU. However, such significance is lost when considering only NICU newborns and is not confirmed by the multivariate analysis. A second theory would be that another factor in NICU babies is involved. One such factor could be the noisy environment of neonatal intensive care units, whose effect on cochlear damage is known and actively addressed by specific indications from the American Academy of Pediatrics. Despite the aforementioned indications, most NICU fail to keep the noise within 45 dB, as recommended [25]. Unfortunately, we do not have the noise level data of our NICU for the whole period to back up this theory. It is important to note that the role of NICU as a confounding factor could be present but pass totally unnoticed in studies on risk factors focused only on the total screened population that do not perform a restriction or multivariate analysis [15], while a lack of significance of some of the traditional risk factors could be noticed in NICU only studies [26]. Although the role of NICU stay is intriguing, and with further data may lead to reconsider the role of other assessed risk factors, caution is required, as this series has a relatively small number of events in terms of hearing loss cases and lacks proper follow-up of hearing-impaired babies.

Our study has some limitations. It has an observational design, and thus, it may be plagued by selection bias. However, it discusses a universal newborn hearing screening program that grants access to all newborns, with virtually no excluded newborns in the covered territory. We reckon that it is possible that a small number of patients referred to our hospital from other facilities might choose to perform the screening at private centers, but to the best of our knowledge, this number is negligible. Risk factors were collected at birth and are limited to the ones that were known at delivery time, thus excluding all those discovered later in life (genetic predisposition, serologic testing on the babies, and definitive diagnosis of syndromes). Some of the anamnestic data were collected through questionnaires, thus being at risk of recall bias, among others. The instrument dedicated to the first stage of screening of NICU babies was changed at the beginning of 2019 with a new model, the Madsen AccuScreen Type 1077 TE/DP, but whether it may have had an impact on referral rates is unknown. To assess the overall percentage of hearing-impaired babies, we used the results of the last available ABR. Some babies might have had middle ear effusion at the time of our diagnostical ABR, and although our common practice includes appropriate treatment and retest in these cases, in some of them, the follow-up ABR might have been missed, or the condition might not have resolved by that time, and thus, some babies with middle ear effusion might have been inappropriately included in the count of permanent hearing losses. The relevant number of dropouts, especially in the diagnostic phase, reduces the number of hearing-impaired babies found and our ability to correlate these missing events with risk factors. Moreover, our screening program has not been audited during the whole period. Strengths of this study include the number of newborns enrolled in a single facility, the uniformity of the testing method, the team of technicians (the same throughout the 9 years included), and the consistent collection of data through the years.

## 5. Conclusions

Our data are largely consistent with the literature, and most results were expected, one relevant exception being the possible role of NICU stay as a confounding factor, the limited number of risk factors confirmed in the multivariate analysis, and the impact this might have on the relevance given to other well-known and assessed risk factors in the future.

## Figures and Tables

**Figure 1 children-09-01362-f001:**
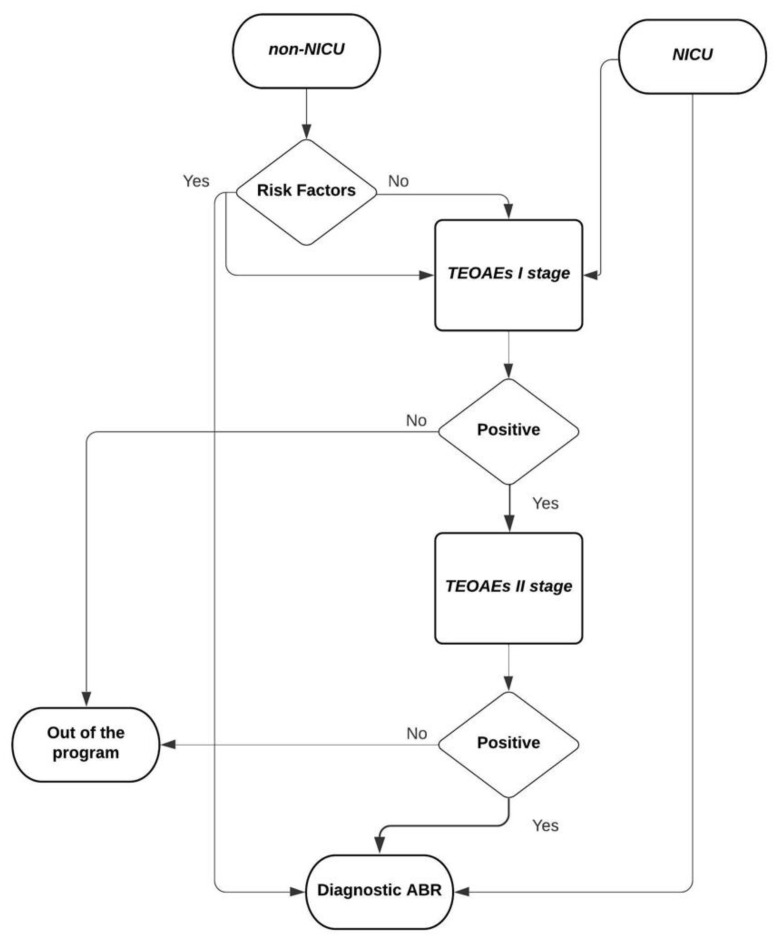
Flowchart of our screening protocol.

**Table 1 children-09-01362-t001:** Refer and dropout rates by year and overall values.

	**Refer after I Stage**		**Refer after II Stage**	
**Year**	**Non-NICU**	**NICU**	**Non-NICU**	**NICU**
2011	62/682 (9.09%)	23/172 (13.37%)	4/682 (0.58%)	4/172 (2.32%)
2012	63/1171 (5.38%)	48/301 (15.95%)	4/1171 (0.34%)	1/301 (0.33%)
2013	72/1133 (6.35%)	32/295 (10.84%)	7/1133 (0.61%)	5/295 (1.69%)
2014	199/1067 (9.37%)	35/261 (13.41%)	1/1067 (0.09%)	9/261 (3.44%)
2015	38/1020 (3.73%)	30/324 (9.26%)	1/1020 (0.09%)	22/324 (9.02%)
2016	36/886 (4.06%)	11/352 (3.12%)	6/886 (0.67%)	3/352 (0.85%)
2017	34/927 (3.67%)	15/301 (4.98%)	8/927 (0.86%)	7/301 (2.32%)
2018	32/949 (3.37%)	10/292 (3.42%)	14/949 (1.47%)	5/292 (1.71%)
2019	112/1116 (10.04%)	26/439 (5.92%)	10/1116 (0.89%)	10/439 (2.27%)
All yrs	648/8951 (7.23%)	230/2737 (8.40%)	55/8951 (0.61%)	66/2737 (2.41%)
	**Dropout II Stage**		**Dropout ABR**	
**Year**	**Non-NICU**	**NICU**	**Non-NICU**	**NICU**
2011	19/62 (30.64%)	8/23 (34.78%)	5/13 (38.46%)	62/172 (36.04%)
2012	16/63 (25.39%)	11/48 (22.91%)	1/54 (1.85%)	95/301 (31.56%)
2013	13/72 (18.05%)	4/31 (12.90%)	7/40 (17.5%)	101/295 (34.23%)
2014	71/100 (71%)	7/35 (20%)	11/44 (25%)	94/261 (36.01%)
2015	22/38 (57.89%)	3/30 (10%)	9/43 (20.93%)	81/324 (25%)
2016	11/36 (30.55%)	6/11 (54.54%)	11/40 (27.5%)	109/352 (30.96%)
2017	15/34 (44.11%)	3/15 (20%)	16/40 (40%)	73/301 (24.25%)
2018	15/32 (46.87%)	1/10 (10%)	6/45 (13.33%)	74/292 (25.34%)
2019	13/112 (11.60%)	1/26 (3.84%)	10/49 (20.4%)	175/439 (39.86%)
All yrs	195/549 (35.51%)	44/229 (19.21%)	76/368 (20.65%)	864/2737 (31.56%)

Note: all babies in the NICU and all babies with risk factors in general receive indication to ABR. Therefore, “dropout ABR” is calculated on all babies that received indication, not only babies with a positive result after the second stage. Refer II stage percentage is calculated on the whole screened population and not on babies tested at the second stage only. Abbreviations: All years (All yrs).

**Table 2 children-09-01362-t002:** Descriptive statistics and comparison of clinical and audiological characteristics of NICU (*n* = 2737) and non-NICU (*n* = 8951) babies. The degree of hearing loss is defined as follows: 0 (26–40 dB HL), 1 (41–55 dB HL), 2 (56–70 dB HL), 3 (71–90 dB HL), and 4 (>90 dB HL). Apgar 0–4/0–6 refers to the apgar score at 1 and 5′. Serological positive tests were performed on mothers, and seldom confirmed on babies. Ototoxic medications refer to aminoglycosides in multiple courses and/or in association with loop diuretics. Percentages are calculated on the total number of newborns of the group, unless otherwise specified. Statistically significant values are marked with an asterisk *.

Clinical Features			
	Non-NICU	NICU	*p*-Value (Log Odds Ratio—95% IC)
Sex	F 4379; (48.92%)	F 1256; (45.88%)	0.005 *
			(−0.122; −0.208 −0.036)
Family history of hearing loss	Yes 299, (3.34%)	Yes 14; (0.51%)	<0.001 *
			(−1.905; −2.443 −1.368)
TORCH	Yes 12; (0.13%)	Yes 100; (3.65%)	<0.001 *
			(3.341; 2.741 3.941)
Low birth weight	Yes 4; (0.04%)	Yes 1503; (54.91%)	<0.001 *
(<1500 g)			(7.910; 6.927 8.893)
Hyperbilirubinemia	Yes 6; (0.06%)	Yes 475; (17.35%)	<0.001 *
			(5.746; 4.940 6.553)
Encephalopathy and meningitis	Yes 0; (0%)	Yes 12; (0.43%)	<0.001 *
			(4.408; 1.581 7.235)
Ototoxic medications	Yes 3; (0.03%)	Yes 75; (2.74%)	<0.001 *
			(4.431; 3.276 5.586)
Opioids	Yes 0; (0.00%)	Yes 13; (0.47%)	<0.001 *
			(4.485; 1.662 7.308)
Connexin 26	Yes 1; (0.01%)	Yes 0; (0%)	0.58
			(0.086; −3.115 3.287)
Apgar 0–4/0–6	Yes 0; (0%)	Yes 255; (9.31%)	<0.001 *
			(7.519; 4.744 10.294)
Syndromes	Yes 14; (0.15%)	Yes 68; (2.48%)	<0.001 *
			(2.789; 2.212 3.366)
Prematurity	Yes 2; (0.02%)	Yes 666; (24.33%)	<0.001 *
			(7.272; 5.883 8.660)
Mechanical ventilation for at least 5 days	Yes 1; (0.01%)	Yes 661; (24.15%)	<0.001 *
			(7.955; 5.993 9.917)
Diabetes	Yes 3; (0.03%)	Yes 74; (2.70%)	<0.001 *
			(4.417; 3.262 5.573)
Craniofacial anomalies	Yes 15; (0.16%)	Yes 27; (0.98%)	<0.001 *
			(1.781; 1.148 2.414)
Cytomegalovirus	Yes 7; (0.07%)	Yes 25; (0.91%)	<0.001 *
			(2.466; 1.627 3.305)
Toxoplasma	Yes 2; (0.02%)	Yes 17; (0.62%)	<0.001 *
			(3.331; 1.865 4.797)
Rubella	Yes 1; (0.01%)	Yes 5; (0.18%)	<0.001 *
			(2.796; 0.694 4.944)
HIV	Yes 2; (0.02%)	Yes 3; (0.11%)	0.053
			(1.591; −0.198 3.381)
HBV	Yes 0; (0.00%)	Yes 2; (0.07%)	0.011 *
			(2.795; −0.242 5.832)
HCV	Yes 0; (0.00%)	Yes 4; (0.14%)	<0.001 *
			(3.383; 0.461 0.306)
HZV	Yes 0; (0.00%)	Yes 1; (0.03%)	0.071
			(2.284; −0.917 5.485)
Total concomitant known audiological risk factors	0:8615; (96.24%)	0:0; (0%)	<0.001 *(-)
	1:331; (3.69%)	1:702; (25.64%)	
	2:5; (0.05%)	2:915; (33.43%)	
	3:0; (0%)	3:636; (23.23%)	
	4:0; (0%)	4:318; (11.61%)	
	5:0; (0%)	5:144; (5.26%)	
	6:0; (0%)	6:20; (0.73%)	
	7:0;(0%)	7:2; (0.07%)	
Audiological features			
	non-NICU	NICU	*p*-value (Log odds ratio—95% IC)
Hearing loss	Yes 35; (0.39%)	Yes 98; (3.58%)	<0.001 *
			(2.247; 1.859 2.635)
Profound bilateral hearing loss	Yes 2; (0.02%)	Yes 2; (0.07%)	0.566
			(0.492; −1.206 2.190)
Degree of hearing loss (worst ear)	0:15; (0.16%)	0:64; (2.33%)	0.055(-)
	1:9; (0.10%)	1:22; (0.80%)	
	2:5; (0.05%)	2:8; (0.29%)	
	3:1; (0.01%)	3:1; (0.03%)	
	4:5; (0.05%)	4:3; (0.11%)	
Unilateral/bilateral hearing loss	Bil 27/35; (77.14%)	Bil 57/98; (58.16%)	0.046 *
			(0.887; 0.002 1.772)
Side in unilateral cases (right–left)	R 6/8; (75%)	R 9/41; (21.95%)	0.002 *
			(−2.398; −4.160 −0.636)
Hearing loss cases with (RF) and without risk factors	RF 14/35; (40%)	RF 98/98; (100%)	<0.001 *
			(5.677; 2.820 8.535)

**Table 3 children-09-01362-t003:** Results of Pearson’s Chi-square test for contingency tables of risk factors and hearing loss status (any degree) in all newborns and in non-NICU and NICU newborns separately. Statistically significant values are marked with an asterisk *. Missing values (no events) and invalid tests are marked with a “-”. HL = hearing loss, NHL = no hearing loss.

Risk Factor	All Newborns HL	All Newborns NHL	*p*-Value	Non-NICU HL	Non-NICU NHL	*p*-Value	NICU HL	NICU NHL	*p*-Value
	133	11,555	(Log Odds Ratio—95% IC)	35	8916	(Log Odds Ratio—95%)	98	2639	(Log Odds Ratio—95% IC)
Sex	F 53; (0.94%)	F 5582; (99.05%)	0.052	F 13; (0.29%)	F 4366; (99.70%)	0.162	F 40; (3.18%)	F 1216; (96.81%)	0.305
			(−0.344; −0.693 0.005)			(−0.485; −1.172 0.202)			(−0.214; −0.624 0.196)
Family history of hearing loss	Yes 13; (9.77%)	Yes 300; (2.59%)	<0.001 *	Yes 13; (37.14%)	Yes 286; (3.2%)	<0.001 *	Yes 0; (0%)	Yes 14; (0.53%)	-
			(1.402; 0.819 1.986)			(2.881; 2.185 3.577)			
TORCH	Yes 3; (2.25%)	Yes 109; (0.94%)	-	Yes 0; (0%)	Yes 12; (0.13%)	-	Yes 3; (3.06%)	Yes 97; (3.67%)	-
Low birth weight (<1500 g)	Yes 62; (46.61%)	Yes 1445; (12.50%)	<0.001 *	Yes 0; (0%)	Yes 4; (0.04%)	-	Yes 62; (63.26%)	Yes 1441; (54.60%)	0.091
			(1.810; 1.465 2.155)						(0.359; −0.059 0.777)
Hyperbilirubinemia	Yes 12; (9.02%)	Yes 469; (4.05%)	0.004 *	Yes 0; (0%)	Yes 6; (0.6%)	-	Yes 12; (12.24%)	Yes 463; (17.54%)	0.174
			(0.852; 0.252 1.452)						(−0.422; −1.034 0.190)
Encephalopathy and meningitis	Yes 1; (0.75%)	Yes 11; (0.09%)	-	-	-	-	Yes 1; (1.02%)	Yes 11; (0.41%)	-
Ototoxic medications	Yes 1; (0.75%)	Yes 77; (0.66%)	-	Yes 0; (0%)	Yes 3; (0.03%)	-	Yes 1; (1.02%)	Yes 74; (2.80%)	-
Opioids	Yes 0; (0%)	Yes 13; (0.11%)	-	-	-	-	Yes 0; (0%)	Yes 13; (0.49%)	-
Connexin 26	Yes 1; (0.75%)	Yes 0; (0%)	-	Yes 1; (2.85%)	Yes 0; (0%)	-	-	-	-
Apgar 0–4/0–6	Yes 15; (11.27%)	Yes 240; (2.07%)	<0.001 *	-	-	-	Yes 15; (15.30%)	Yes 240; (9.09%)	0.038 *
			(1.791; 1.238 2.343)						(0.591; 0.026 1.157)
Syndromes	Yes 10; (7.51%)	Yes 72; (0.62%)	<0.001 *	Yes 1; (2.85%)	Yes 13; (0.14%)	-	Yes 9; (9.18%)	Yes 59; (2.23%)	<0.001 *
			(2.562; 1.877 3.247)						(1.487; 0.754 2.219)
Prematurity	Yes 23;(17.29%)	Yes 645; (5.58%)	<0.001 *	Yes 0; (0%)	Yes 2; (0.02%)	-	Yes 23; (23.46%)	Yes 643; (24.36%)	0.839
			(1.263; 0.807 1.720)						(−0.049; −0.525 0.426)
NICU	Yes 98; (73.68%)	Yes 2639; (22.83%)	<0.001 *	-	-	-	-	-	-
			(2.247; 1.859 2.666)						
Mechanical ventilation for at least 5 days	Yes 28; (21.05%)	Yes 634; (5.48%)	<0.001 *	Yes 0; (0%)	Yes 1; (0.01%)	-	Yes 28; (28.57%)	Yes 633; (23.98%)	0.298
			(1.525; 1.100 1.949)						(0.237; −0.210 0.684)
Diabetes	Yes 0; (0%)	Yes 77; (0.66%)	-	Yes 0; (0%)	Yes 3; (0.03%)	-	Yes 0; (0%)	Yes 74; (2.80%)	-
Craniofacial anomalies	Yes 9; (6.76%)	Yes 33; (0.28%)	<0.001 *	Yes 3; (8.57%)	Yes 12; (0.13%)	-	Yes 6; (6.12%)	Yes 21; (0.79%)	<0.001 *
			(3.232; 2.474 3.990)						(2.096; 1.165 3.026)
Cytomegalovirus	Yes 2; (1.50%)	Yes 30; (0.25%)	-	Yes 0; (0%)	Yes 7; (0.07%)	-	Yes 2; (2.04%)	Yes 23; (0.87%)	-
Toxoplasma	Yes 0; (0%)	Yes 19; (0.16%)	-	Yes 0; (0%)	Yes 2; (0.02%)	-	Yes 0; (0%)	Yes 17; (0.64%)	-
Rubella	Yes 0; (0%)	Yes 6; (0.05%)	-	Yes (0; 0%)	Yes 1; (0.01%)	-	Yes 0; (0%)	Yes 5; (0.18%)	-
HIV	Yes 1; (0.75%)	Yes 4; (0.03%)	-	Yes 0; (0)	Yes 2; (0.02%)	-	Yes 1; (1.02%)	Yes 2; (0.07%)	-
HBV	Yes 0; (0%)	Yes 2; (0.01%)	-	-	-	-	Yes 0; (0%)	Yes 2; (0.07%)	-
HCV	Yes 0; (0%)	Yes 4; (0.03%)	-	-	-	-	Yes 0; (0%)	Yes 4; (0.15%)	-
HZV	Yes 0; (0%)	Yes 1; (0.008%)	-	-	-	-	Yes 0; (0%)	Yes 1; (0.03%)	-

**Table 4 children-09-01362-t004:** Logistic regression analysis to assess the relationship between clinical characteristics and hearing loss. Statistically significant values are marked with an asterisk *. OR = odds ratio.

	Univariate Analysis		Multivariate Analysis	
	OR (95% CI)	*p*-Value	OR (95% CI)	*p*-Value
Males	0.71 (0.50–1.01)	0.05 *	-	-
Prematurity	3.54 (2.24–5.58)	<0.001 *	1.01 (0.31–3.32)	1.38
Family history of hearing loss	4.06 (2.27–7.29)	<0.001 *	17.55 (5.04–61.09)	<0.001 *
TORCH	2.42 (0.76–7.73)	0.14	-	-
Low birth weight (<1500 g)	6.11 (4.33–8.63)	<0.001 *	3.19 (1.04–9.80)	0.04 *
Hyperbilirubinemia	2.34 (1.29–4.27)	0.05 *	0.83 (0.25–2.79)	1.16
Encephalopathy and meningitis	7.95 (1.02–62.02)	0.05 *	-	-
Ototoxic medications	1.13 (0.16–8.18)	1.3	-	-
Mechanical ventilation for at least 5 days	4.59 (3.00–7.02)	<0.001 *	1.78 (0.56–5.66)	0.33
Apgar 0–4/0–6	5.99 (3.45–10.41)	<0.001 *	2.51 (0.72–8.69)	0.15
NICU	9.46 (6.42–13.95)	<0.001 *	11.45 (3.29–39.89)	<0.001 *
Syndromes	12.97 (6.54–25.72)	<0.001 *	3.42 (0.77–15.21)	0.11
Total concomitant known audiological risk factors	1.78 (1.62–1.95)	<0.001 *	0.67 (0.23–1.92)	0.45
Cytomegalovirus	5.87 (1.39–24.80)	0.02 *	4.05 (0.74–21.98)	0.11
Toxoplasma	-	-	-	-
Rubella	-	-	-	-
HIV	21.88 (2.43–197.05)	0.06	13.20 (1.22–143.10)	0.03 *
HBV	-	-	-	-
HCV	-	-	-	-
HZV	-	-	-	-
Opioids	-	-	-	-
Diabetes	-	-	-	-
Craniofacial anomalies	25.34 (11.88–54.08)	<0.001 *	9.62 (2.97–31.13)	<0.001 *

## Data Availability

The data that support the findings of this study are available upon reasonable request from the corresponding author.

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
