# Peer review of "Audiological Risk Factors, Referral Rates and Dropouts: 9 Years of Universal Newborn Hearing Screening in North Sardinia"

_children, 2022, doi:10.3390/children9091362_

Round 1

Reviewer 1 Report

-       Abstract: replace ‘OAU Sassari’ into ‘University Hospital Sassari (Italy)’

-       Abstract: can you state ‘had some degree of hearing loss’ based on screening alone? It seems to me that a confirmation is necessary, because other UNHS studies see about 20% of referred patients not exhibiting any hearing loss.

-       Introduction: can you be more specific on the etiology of hearing loss? A sentence like ’50-60% of cases are associated with defined genetic or non-genetic factors…’ raises more questions than answers.

-       Methods: please write every abbreviation in long when encountered for the first time (NHL).

-       Methods: it is unclear which children are born in your center and which are referred. I suppose these are not all the children of Sardinia? Is a referral bias possible (mainly children at risk are referred, all potentially NICU babies are born in your center, non-NICU babies at your center might be more at risk)? This is important to interpret the prevalence.

-       Methods: why would you screen NICU and non-NICU babies with TEOAEs as a diagnostic ABR is mandatory considering the protocol?

-       Methods: please provide information about JASP (company, country).

-       Methods: was this study approved by a local Ethics Committee? You cannot just report data retrospectively, even anonymized. I am afraid that the authors performed the study without approval, as they deleted the ‘institutional review board statement’ proposed by the MDPI template.

-       Results: The results are ambiguously presented. I believe you should start by repeating Figure 1 and providing the number of babies in each situation. This would clarify a lot. You cannot say in the second sentence of the results how much children have hearing loss (based on ABR or on screening?) because you did not explain the trajectory. You should provide absolute number + percentages for all calculations. There should be a clear path in which the reader can follow where all numbers come from.

-       Results: Is it correct that 52% of the non-NICU newborns eventually exhibiting hearing loss, were not picked up by screening? How can you explain this taking into account the estimated sensitivity?

-       Results: How did you deal with a mild and temporary hearing loss on ABR due to middle ear effusion? Are these included in the numbers? This is important as you compare with 1-3/1000, which is the number for permanent congenital hearing loss.

-       Results: Figure 2 cannot be interpreted because of too low quality.

-       Results: Table 1: how did you define profound hearing loss (category 4)? Is the degree of hearing loss for the best or the worst ear?

-       Results: the tables need some serious improvement in view of formatting (e.g. width of the columns so everything is easy to read)

-       Results: a Pearson chi square cannot be used if one of the cells is <5.

-       Results: in your multivariate analysis, did you correct for multicollinearity? 

-       Discussion: the sentence ‘The most referrals to audiological centres are usually due to false positive results’ is wrong as it implies that the majority is false positive.

-       Discussion: can you provide a reference for temporary hearing loss usually resolving with hours (?) or days?

-       Discussion: the explanation of risk factors being not adequately represented, presumably due to lack of diagnostics GJB2) or the association with delayed onset hearing loss (cCMV) is not valid. Risk factors (e.g. family history, infection during pregnancy,…) are different than etiological results (GJB2, confirmed cCMV infection), so the lack of etiological results cannot explain the skewed distribution of hearing loss severity. Moreover, the delayed onset of cCMV-related hearing loss is much lower than congenital involvement.

-       Discussion: can you provide a reference for NICU noise causing cochlear damage?

Author Response

Dear Reviewer, thank you for the valuable inputs provided. Here follows a list of our answers and corrections.

Abstract: replace ‘OAU Sassari’ into ‘University Hospital Sassari (Italy)’

  • Thank you, changed

Abstract: can you state ‘had some degree of hearing loss’ based on screening alone? It seems to me that a confirmation is necessary, because other UNHS studies see about 20% of referred patients not exhibiting any hearing loss.

  • The number reflects the percentage of babies with confirmed hearing loss (diagnostic ABR)

Introduction: can you be more specific on the etiology of hearing loss? A sentence like ’50-60% of cases are associated with defined genetic or non-genetic factors…’ raises more questions than answers.

  • Thank you, we expanded the paragraph a bit for better clarity.

Methods: please write every abbreviation in long when encountered for the first time (NHL).

  • Thanks, we checked the other abbreviations and we eliminated NHL altogether to avoid confusion with the same abbreviation used in table 2 with a different meaning.

Methods: it is unclear which children are born in your center and which are referred. I suppose these are not all the children of Sardinia? Is a referral bias possible (mainly children at risk are referred, all potentially NICU babies are born in your center, non-NICU babies at your center might be more at risk)? This is important to interpret the prevalence.

  • We tried to better clarify in the text that we perform the UNHS for all the babies born in North Sardinia (Sassari Province), all the babies are referred to our hospital. We reckon that there might be a source of bias related to the fact that some parents referred from other hospitals might prefer to go to private services instead of ours (public and free), but from the (limited) numbers in our possession this seems to be a relatively small percentage. We discussed this as a potential source of bias in the appropriate section.

Methods: why would you screen NICU and non-NICU babies with TEOAEs as a diagnostic ABR is mandatory considering the protocol?

  • Thank you for your observation, this is definitely a point that needs clarification. According to JCIH 2007 ABR is the recommended screening technique in a NICU setting (mainly to avoid missing auditory neuropathies). We do not dispose of AABR technology, thus our only option is to offer a diagnostic ABR to all babies, usually performed after discharge. We still perform TEOAEs on all babies, that at this point would seem redundant, because this still allow us to have a first contact with parents (before discharge), to counsel them, to collect informations on risk factors other than NICU stay and to identify early babies with a higher risk (refers). We agree that this might not be the best nor the most efficient choice. JCIH 2007 also recommends a personalized follow up plan for babies with risk factors. It is our policy to perform a first ABR within 3 month and to plan subsequent follow ups for well born babies that have known risk factors, an information that is collected at the time of the first screening test.

Methods: please provide information about JASP (company, country).

  • We added in text information about the software. It is an open source statistical software supported by the University of Amsterdam and many other funding sources: https://jasp-stats.org/sponsors

Methods: was this study approved by a local Ethics Committee? You cannot just report data retrospectively, even anonymized. I am afraid that the authors performed the study without approval, as they deleted the ‘institutional review board statement’ proposed by the MDPI template.

  • We have discussed the matter with the Editor and provided the relevant documentation, We are waiting for their decision and ready to provide further documentation if needed.

Results: The results are ambiguously presented. I believe you should start by repeating Figure 1 and providing the number of babies in each situation. This would clarify a lot. You cannot say in the second sentence of the results how much children have hearing loss (based on ABR or on screening?) because you did not explain the trajectory. You should provide absolute number + percentages for all calculations. There should be a clear path in which the reader can follow where all numbers come from.

  • Thank you, we added the information in the text, originally we didn’t include in the text too many data available on tables because we thought it would look redundant and repetitive, but of course clarity is paramount, and in fact some data were missing from both text and tables. Hearing loss status was assessed based on the results of diagnostic ABR, this has also been clarified in the text.

Results: Is it correct that 52% of the non-NICU newborns eventually exhibiting hearing loss, were not picked up by screening? How can you explain this taking into account the estimated sensitivity?

  • From the text: “52.04% (51/98) of NICU newborns, and 14.28% (5/35) of non-NICU newborns with hearing impairment had passed the first stage of screening and were referred to ABR due to risk factors”. Yes, the number of NICU babies that passed the first stage of screening but had pathological diagnostic ABR is particularly high, and higher than the non-NICU counterpart. We think that the only possible interpretation is that these babies had an etiologic factor that caused a delayed hearing loss, or the etiologic factor itself occurred during the time frame between the TEOAEs and the diagnostic ABR. Both are possible, especially the second considering the high number of potential etiologic factor related to a NICU stay. We added this consideration in the text.

Results: How did you deal with a mild and temporary hearing loss on ABR due to middle ear effusion? Are these included in the numbers? This is important as you compare with 1-3/1000, which is the number for permanent congenital hearing loss.

  • Babies with middle ear effusion were treated with topical treatments as per common practice, and followed up with another ABR. Temporary hearing losses are not included in our report.

Results: Figure 2 cannot be interpreted because of too low quality.

  • Thanks, the quality was better but apparently the image was further compressed during the upload, so we converted the figure to a table anyway for better clarity.

Results: Table 1: how did you define profound hearing loss (category 4)? Is the degree of hearing loss for the best or the worst ear?

  • We included an explanation in the table description: “The degree of hearing loss is defined as follows: 0 (26-40 dB HL), 1 (41-55 dB HL), 2 (56-70 dB HL), 3 (71-90 dB HL), 4 (>90 dB HL).” The degree is in relation to the worst ear (specified in the table). We reckon that this might be debatable, let us know if you think that another approach could be better.

Results: the tables need some serious improvement in view of formatting (e.g. width of the columns so everything is easy to read)

  • Thank you, the tables were admittedly unfit for the journal’s format, so we edited them.

Results: a Pearson chi square cannot be used if one of the cells is <5.

  • Thank you for highlighting that, also fisher’s exact test can’t be used due to the large numbers of babies. We edited the table and the text to reflect the fact that some of the tests were not valid

Results: in your multivariate analysis, did you correct for multicollinearity?

  • We thank the Reviewer for having raised the present question. Yes, we accounted for multi-collinearity.

Discussion: the sentence ‘The most referrals to audiological centres are usually due to false positive results’ is wrong as it implies that the majority is false positive.

  • We agree that the sentence did not reflect the reality of things, we changed it to better convey our point (depending on the protocol a non negligible number of referral are due to false positive tests)

Discussion: can you provide a reference for temporary hearing loss usually resolving with hours (?) or days?

  • We added a relevant citation, and it is important to stress that we are mainly referring to the presence of meconium

Discussion: the explanation of risk factors being not adequately represented, presumably due to lack of diagnostics GJB2) or the association with delayed onset hearing loss (cCMV) is not valid. Risk factors (e.g. family history, infection during pregnancy,…) are different than etiological results (GJB2, confirmed cCMV infection), so the lack of etiological results cannot explain the skewed distribution of hearing loss severity. Moreover, the delayed onset of cCMV-related hearing loss is much lower than congenital involvement.

  • Agreed, this was an invalid interpretation of our data, we edited it out.

Discussion: can you provide a reference for NICU noise causing cochlear damage?

  • The potential impact of NICU noise on cochlear damage is discussed in the article cited after the next sentence: Committee on Environmental Health; Noise: A Hazard for the Fetus and Newborn. Pediatrics October 1997; 100 (4): 724–727. 10.1542/peds.100.4.724.

Reviewer 2 Report

Title: Audiological risk factors, referral rates and dropouts: 9 years of 2 Universal Newborn Hearing Screening in North Sardinia

In my opinion the article is well prepared.  However, some details that could be improved:

1)      The acronyms should be defined the first time they appear in the text:

Line 55: TEOAEs, DPOAEs, ABRs and AABRs

Line 159: BAHA

2)      Line 107: The reference of JCIH, at this point in the text it should be an earlier version of the JCIH, since the said reference was only updated at the end of 2019, and the study was made between 2011-2019.

3)      Figure 2 has poor quality and becomes unreadable. Data is not received.

4)      The tables should also be better organized

I hope I have helped improve your article.

The best regards

Author Response

Dear Reviewer, thank you for the valuable inputs provided. Here follows a list of our answers and corrections.

In my opinion the article is well prepared.  However, some details that could be improved:

The acronyms should be defined the first time they appear in the text:

Line 55: TEOAEs, DPOAEs, ABRs and AABRs

Line 159: BAHA

  • Thank you, we added the proper definitions

 Line 107: The reference of JCIH, at this point in the text it should be an earlier version of the JCIH, since the said reference was only updated at the end of 2019, and the study was made between 2011-2019.

  • Thank you, we changed the citation.

Figure 2 has poor quality and becomes unreadable. Data is not received.

  • The figure was severely compressed during upload, but it was probably ill conceived from the beginning, so we decided to use a table instead to provide the data (new table 1).

The tables should also be better organized

  • We agree, they do not fit the format of the journal, we changed them accordingly

Round 2

Reviewer 1 Report

I would like to thank the authors for their effort in making the manuscript more comprehensible. However, I still have some remaining points to clarify:

1) To me, the main reason to explain the NICU babies passing their OAE screening but failing their ABR test, is not a delayed onset of hearing loss, but is the presence of auditory neuropathy spectrum disorder (ANSD). This is particularly high in the NICU population, probably due to delayed maturation of the auditory tract, and is not picked up by OAE screening.

2) I do not think your answer about my middle ear effusion question is correct. In no way you described a separate path or exclusion of temporary hearing loss due to middle ear effusion. This pathology can give failed OAE screening and failed ABR testing (mild-moderate hearing loss), so according to your protocol these are included. If not, you should state this in the methods section.

3) Grammar. Some examples are provided below.

Line 45: definite instead of difinite, cannot instead of can't

Line 47: ototoxic medications instead of medications

Line 62: brainstem instead of breainstem

Author Response

Dear Reviewer, thank you once more for the valuable inputs provided, we appreciated them and think that the paper greatly benefited from the revisions. Here follows a list of our answers and corrections.

I would like to thank the authors for their effort in making the manuscript more comprehensible. However, I still have some remaining points to clarify:

1) To me, the main reason to explain the NICU babies passing their OAE screening but failing their ABR test, is not a delayed onset of hearing loss, but is the presence of auditory neuropathy spectrum disorder (ANSD). This is particularly high in the NICU population, probably due to delayed maturation of the auditory tract, and is not picked up by OAE screening.

  • Thank you for this comment. We fully agree, this is for sure one of the main factors to consider and was not stressed enough in the text and was completely missing in our previous response to your comment. We clarified this in the text in two different points of the discussion.

2) I do not think your answer about my middle ear effusion question is correct. In no way you described a separate path or exclusion of temporary hearing loss due to middle ear effusion. This pathology can give failed OAE screening and failed ABR testing (mild-moderate hearing loss), so according to your protocol these are included. If not, you should state this in the methods section.

  • Thank you, this is indeed not clarified in the text. Babies that do not pass the first ABR are usually tested at least two more times before the 6th month of age, and if middle ear effusion is suspected (after the first failed ABR babies are evaluated by an audiologist and a tympanogram is performed) they are treated accordingly and re-tested. To perform our analysis we considered the last diagnostic ABR available in our database within the 6th month. We agree that it is possible that some cases of slight and mild hearing loss due to middle ear effusion might have not been resolved within this time frame and might have been included in the analysis. We clarified this both in the material and methods and in the limitations section.

3) Grammar. Some examples are provided below.

Line 45: definite instead of difinite, cannot instead of can't

Line 47: ototoxic medications instead of medications

Line 62: brainstem instead of breainstem

  • Thank you, we edited all the typos we were able to find manually and with appropriate softwares.
